# DiffRGenNet: Difference-aware Medical Report Generation

**MinghaoBian**[1,2]                                          BIANMINGHAO@MAIL.USTC.EDU.CN
**KunZhang**[1,2]                                                    KKZHANG@USTC.EDU.CN
**DexinZhao**[1,2]                                          DEXINZHAO@MAIL.USTC.EDU.CN
**S.KevinZhou**[*1,2,3,4]                                        SKEVINZHOU@USTC.EDU.CN

[1] *School of Biomedical Engineering, Division of Life Sciences and Medicine, University of Science and Technology of China (USTC)Hefei Anhui, 230026, China*

[2] *Center for Medical Imaging, Robotics, Analytic Computing  Learning (MIRACLE), Suzhou Institute for Advance Research, USTC, Suzhou Jiangsu, 215123, China*

[3] *Jingsu Provincial Key Laboratory of Multimodal Digital Twin Technology, Suzhou Jiangsu, 215123, China*

[4] *State Key Laboratory of Precision and Intelligent Chemistry, USTC, Hefei Anhui 230026, China*

**Editors:** Accepted for publication at MIDL 2025

## Abstract

Medical report generation is a critical task in healthcare, aiming to automatically produce accurate diagnostic reports from medical images, thereby alleviating the burden on radiologists. However, due to the high similarity among medical images of the same anatomical region and the substantial variations captured from the same region across different time points for individual patients, capturing these differences poses a significant challenge. We propose a **Diff**erence-aware **R**eport **Gen**eration **Net**work (DiffRGenNet), which retrieves similar reports through image search, identifies differences using the Feature Diff module, and dynamically orchestrates global and local dependencies via the FlexiRoute Aggregation Module to determine the optimal routing path for each sample, selecting the most suitable report to describe the variations and connections. Finally, by leveraging the consistency of classification information and the discrepancy information from the diff module, DiffR-GenNet enhances the ability to learn differences in rare diseases, generating more precise reports. Experiments demonstrate that DiffRGenNet outperforms existing methods on the MIMIC-CXR and IU X-Ray datasets, confirming its effectiveness and potential.

**Keywords:** Report Generation, Multimodal Learning.

## 1. Introduction

Radiological reports serve as critical foundations for clinical diagnosis and treatment based on medical imaging. In recent years, the automatic generation of radiological reports using deep learning has gained popularity. However, most existing methods focus on generating reports for diseases themselves(Chen et al., 2020; Li et al., 2024) while neglecting individual variability and nuanced features among different conditions. For instance, the subtle differences between Example 1 and Example 2 in Figure 1 make it challenging to visually discern their respective disease states, leading to semantic errors and potentially misleading medical guidance if reports are generated directly from such images. Moreover, in clinical practice, there are multiple X-rays from the same patient at different time points, as well

---

* Zhou is the corresponding author.

Figure 1: Example 1 illustrates the comparison among the majority of normal reports, while Example 2 demonstrates the contrast between abnormal conditions of a patient across different time periods.The red font highlights the key sections in the report.

as a large number of X-rays capturing the same anatomical region across different patients. Therefore, it is key to capture these fine-grained differences to generate precise and detailed reports.

Image difference captioning involves describing the differences between pairs of similar images using natural language. Recent research has explored how to model reliable representations of changes under varying viewpoints (Park et al., 2019; Vo et al., 2019; Huang et al., 2021). In contrast, medical imaging does not need to account for viewpoint variations, focusing solely on the differences between images. Researchers have developed methods to improve the accuracy and comprehensive of medical report generation (MRG), such as leveraging knowledge graphs (Liu et al., 2021; Xiang et al., 2024), and integrating large language models (LLMs) (Liu et al., 2024; Chen et al., 2024b) for prompt generation (Jin et al., 2024), designing auxiliary enhancement modules to improve generation outcomes. Some works frame MRG as an retrieval problem (Endo et al., 2021; Tao et al., 2024), assisting generation by extracting top-$k$ features most similar to the input image. Our method also introduces retrieval techniques.However, features may include both relevant and irrelevant information and the irrelevant part interferes with the representation of image features.Our retrieval approach builds upon previous methods by utilizing the retrieved content in a more fine-grained manner. We differentially handle the similar and significantly different aspects of the retrieved content, thereby enhancing the learned knowledge.

Despite some progress, there is still limited research focused on extracting *differential changes between radiological images and reports*. Furthermore, the inherent imbalance in disease distribution exacerbates the challenge of capturing differential variations, as rare diseases are underrepresented in training data, hindering the model's ability to reliably identify characteristic changes in these conditions. Existing models predominantly trained on positive samples, exhibit a bias toward common diseases and fail to effectively discern subtle variations in rare diseases across different time points or patients. Additionally, most current models rely on attention mechanisms within Transformer architectures, which struggle to dynamically balance global and local dependencies and often result in an inability

to simultaneously capture global structures and local details, thereby impeding the selection of optimal reports to describe differential changes and correlations in retrieval.

To fill this gap, we design a novel network, **Diff**erence-aware **R**eport **Gen**eration **Net**work (**DiffRGenNet**), to leverage both differential and similar information across reports for generating more accurate and reliable medical reports through finer-grained global and local feature extraction. Specifically, building upon an encoder-decoder framework, DiffRGenNet retrieves $K$ reports most similar to the input image and employs the Flexible Aggregation Module (FAM) to dynamically select the optimal report for describing differential changes and correlations. The FAM module captures both global and local features, distinguishing between similar features (extracted via a classification branch to identify disease information) and differential features (extracted via a dedicated diff module to highlight variations between images). By contrasting positive and negative samples, the model aligns closely with ground truth while minimizing noise, thereby enhancing its ability to capture subtle variations in rare diseases. Extensive experiments on two MRG benchmarks demonstrate the effectiveness of our approach, achieving state-of-the-art results. Our contributions are summarized as follows: (i) We propose a novel network framework, DiffRGenNet, which utilizes contrastive learning with feature-differential negative samples to effectively capture nuanced variations and generate more fine-grained medical reports. (ii) We design the Flexible Aggregation Module (FAM) to adaptively capture the most relevant global and local features for describing differential variations and their correlations. Further, we introduce a specific module to focus on disease-related changes. (iii) We demonstrate the superiority of DiffRGenNet through evaluations on two widely recognized benchmarks, achieving state-of-the-art (SOTA) performances on both datasets.

## 2. Method

Given a radiological image $I$, the model is required to generate a descriptive radiological report $\tilde{R} = \{r_1, r_2, \ldots, r_{N_R}\}$, where $r_i$ represents a token in the report and $N_R$ is the length of the report. The recursive generation process can be formulated as $P(\tilde{R}|I) = \prod_{t=1}^{T} p(r_{t+1}|r_1, r_2, \ldots, r_t, I)$. DiffRGenNet estimates $P(\tilde{R}|I)$ via a network, which primarily comprises three modules as in Figure 2: (i) the Feature Difference Module, discussed in Section 2.1;(ii)the FlexiRoute Aggregation Module, detailed in Section 2.2; and (iii) the Neg-Pos Matching, outlined in Section 2.3.

### 2.1. Feature Difference Module

In MRG, each report meticulously describes the affected regions and associated symptoms of a patient, derived from identifying and characterizing abnormal areas in medical images. The differential metric mechanism (Tu et al., 2023b,a) enhances the model's sensitivity to input features, enabling it to more effectively capture critical clinical information within the images, thereby improving the accuracy and reliability of the generated reports.

To effectively quantify these differences, We retrieve the top (N) reports that are most similar to the report paired with the image, and then use Feature Difference Module to capture the differences between the image and these similar reports.This module compares image and text embeddings using three distinct metrics: L2 distance, cosine similarity, and dot product similarity. These metrics facilitate the quantification of both similarities and

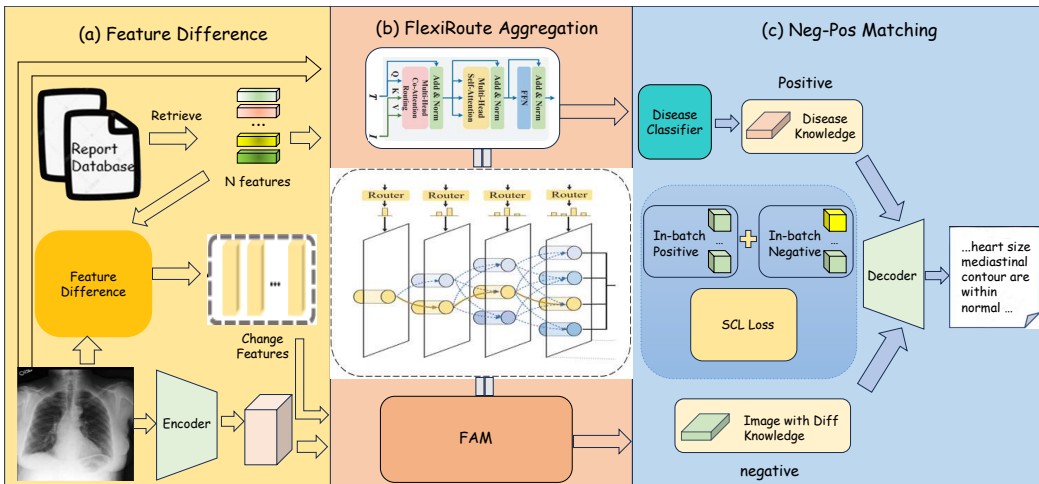

Figure 2: The architecture of DiffRGenNet. It integrates a feature difference module, a FlexiRoute aggregation module (FAM), and contrastive learning to generate more fine-grained and precise medical reports.

differences between input features and reference embeddings from multiple perspectives, enhancing the robustness of the model. The L2 distance, also known as the Euclidean distance, represents the straight-line distance between two vectors. In this task, the L2 distance is used to measure the difference between the input feature $Z_k$ and the average embedding $F$: $dif_{L2} = \|Z_k - F\|_2$ Cosine similarity measures the similarity in direction between two vectors, regardless of their magnitudes. Here, we use cosine similarity to quantify the similarity between the input feature and the average embedding: $dif_{cos} = \frac{z_k F}{\|z_k\|_2 \|F\|_2} \in \mathbb{R}^{N \times 1}$ Dot product similarity measures the inner product of two vectors, reflecting the degree of overlap in the same direction: $dif_{dot} = Z_k F$ These vectors are concatenated to obtain:

$$dif = \text{MLP}(\text{concat}(dif_{L2}, dif_{cos}, dif_{dot})) \in \mathbb{R}^{N \times d_h}. \tag{1}$$

## 2.2. FlexiRoute Aggregation Module

Transformer, renowned for its exceptional capability in modeling global dependencies, has been widely adopted in medical report generation. However, the challenge of dynamically balancing global and local dependencies within Transformer architectures remains unresolved. To address this, we propose the FlexiRoute Aggregation Module (FAM) as in Figure 2, which introduces a routing mechanism with a varying attention spanning at each layer of the vision Transformer. This module dynamically computes attention weights based on the output of previous step, enabling the generation of an optimal routing path for each sample. This approach significantly enhances the retrieval process by facilitating the selection of the most suitable report, thereby improving the overall system performance. The FAM module supports inputs from both image and text modalities. In DiffRGenNet, similar features are selected by inputting the image and the retrieved N features. For difference features, they are selected by inputting the image and the Change Features.

In the FlexiRoute Aggregation Module, feature embeddings are processed through multiple Dynamic Routing Attention (DRA) layers, computed as follows:

$$Z_k = DRA(Z_{k-1}, F), k \in [1, K], \tag{2}$$

where $Z_k$ represents the output of the $k$-th DRA layer, $Z_0 = Z$ denotes the input to the first layer, $K$ is the maximum index of DRA layers, and the output $Z_k$ of the final DRA layer constitutes the ultimate routed features. In contrast to prior dynamic methods such as TRAR (Zhou et al., 2021), which perform routing on a single feature's attention grid, our DRA layers route hierarchical co-attention across both image and text features, conditioned on the specific input. Each DRA layer consists of a Multi-Head Co-Attention Routing (MH-CAR) module, a Multi-Head Self-Attention (MHA) module, and a Feed-Forward Network (FFN), with each module followed by a residual connection and a layer normalization (LN). The $k$-th DRA layer can be expressed as:

$$
\begin{aligned}
Z_{k-1}^r &= \text{LN}(\text{MHCAR}_k(Z_{k-1}, F) + Z_{k-1}), \\
Z_{k-1}^a &= \text{LN}(\text{MHA}_k(Z_{k-1}^r) + Z_{k-1}^r), \\
Z_k &= \text{LN}(\text{FFN}_k(Z_{k-1}^a) + Z_{k-1}^a),
\end{aligned}
\tag{3}
$$

where $k \in [1, K]$ denotes the index of the DRA layer, $Z_k \in \mathbb{R}^{n \times d_t}$ represents the output of the $k$-th DRA layer, and $Z_{k-1}^r$ and $Z_{k-1}^a$ are the outputs of the MHCAR module and the MHA module, respectively. In the $k$-th DRA layer, the MHCAR module performs an $h$-head attention function, computing the hidden dimension $d_h$ (where $d_h = d_t/h$) in parallel for each head. The results from these heads are concatenated and then projected to produce the final output of the MHCAR module. This process can be formulated as:

$$
\text{MHCAR}_k(Z_{k-1}, F) = \text{concat}([head_i^k]_{i=1}^h)O_T^k,
\tag{4}
$$

where $\text{concat}(\cdot)$ denotes the concatenation operation, $O_T^K \in \mathbb{R}^{d_t \times d_t}$ is the projection matrix, and each head $head_i^k \in \mathbb{R}^{n \times d_h}$ is computed by the Co-Attention Routing (CAR) function, formulated as:

$$
head_i^k = \text{CAR}_i^k(Z_{k-1}, F) = \sum_{j=0}^{p_{k-1}} \alpha_j^k CA_{i,j}^k(Q_{i,j,k}, K_{i,j,k}, V_{i,j}^k, A^j) = \sum_{j=0}^{p_k-1} \alpha_j^k \sigma\left(\frac{Q_{i,j,k}K_{i,j,k}}{\sqrt{d_h}} \otimes A^j\right) V_{i,j}^k,
\tag{5}
$$

where $\sigma(\cdot)$ is the softmax function, $\alpha_j^k$ is the routing probability weight for the $j$-th co-attention function, $A^j$ is a co-attention mask between the two features, and $Q_{i,j,k}$ and $K_{i,j,k}$ are the attention matrices between the two features for the $head_i^k$. Here, $Q_{i,j,k} = Z_{k-1}W_{i,j,k}^Q$, $K_{i,j,k} = FW_{i,j,k}^K$, and $V_{i,j,k} = FW_{i,j,k}^V$, where $W_{i,j,k}^Q \in \mathbb{R}^{d_f \times d_h}$, $W_{i,j,k}^K \in \mathbb{R}^{d_f \times d_h}$, and $W_{i,j,k}^V \in \mathbb{R}^{d_f \times d_h}$ are parameter matrices, and $\otimes$ denotes element-wise matrix multiplication.

We describe the construction of the co-attention mask matrix $A^j$, which restricts the relevant regions that image features can attend to within the co-attention function. Using an $s$-order sliding window, a patch of size $(2s + 1) \times (2s + 1)$ traverses each block of the image, generating a mask vector $v_l^s \in \mathbb{R}^m$ (where $l \in [1, m]$). The matrix $A^s$ is constructed by cyclically stacking the vector $v_l^s$ $n$ times (where $n$ is the token length):

$$
A^s = [v_l^s, v_l^s, ..., v_l^s] \in R^{nm}.
\tag{6}
$$

Specifically, $A^0$ is an empty mask matrix, i.e., a matrix filled with ones, allowing words or the global token [CLS] to attend to the entire image. To progressively model the consistency between different feature pairs, we design a hierarchical co-attention mechanism

by incrementally increasing the number of DAR layers, thereby diversifying the types of co-attention masks. In the $k$-th DAR layer, the set of co-attention mask matrices that the router can route is defined as: $G_k = [A^0, A^1, ..., A^{p_k-1}]$, where $p_k$ denotes the number of mask matrices in the $k$-th DAR layer. The routing probabilities $\alpha_k = [\alpha_k^0, \alpha_k^1, \ldots, \alpha_k^{p_k-1}]$ for the $k$-th DAR layer can be obtained by the router based on the input conditions. The calculation formula is as follows:

$$\alpha_k = \sigma_g(\text{MLP}(\text{APool}(F))) \in R^{p_k}, \tag{7}$$

where $\sigma_g(\cdot)$ is the Gumbel-Softmax function with temperature $t$, $\text{APool}(\cdot)$ denotes the 1D adaptive average pooling over all patch embeddings in the image, MLP is a two-layer multi-layer perceptron with hidden dimension $d_m$.

### 2.3. Neg-Pos Matching

Unlike most existing studies which typically reinforce high-relevance segments by associating cross-modal shared semantics while weakening or even ignoring the impact of mismatched segments,our work transcends the limitation of solely focusing on enhancing attention to matched segments (Zhang et al., 2023, 2022). We employ supervised contrastive learning (SCL) to simultaneously align both similar and dissimilar segments, thereby more comprehensively capturing cross-modal semantic relationships.

**SCL Loss**. The objective of SCL is to learn useful representations of data by maximizing the similarity between positive samples while minimizing the similarity between negative samples. In SCL, the model learns representations by comparing pairs of samples (anchor samples, positive samples, and negative samples). Specifically, given an anchor sample, the goal is to make it more similar to positive samples and less similar to negative samples. This is achieved by computing similarity scores between samples and applying a variant of the contrastive loss function. In this task, we partition the representations in each batch into multiple subsets based on whether they share the same sample label. Then, for each subset, the representations within the subset serve as positive samples, while those from other subsets act as negative samples.

$$\mathcal{L}_{SCL} = -\frac{1}{N} \sum_{i=1}^{N} \log \left( \frac{e^{f(x_i, x_i^+)}}{\sum_{j=1}^{K} e^{f\left(x_i, x_j^-\right)}} \right), \tag{8}$$

where $N$ is the number of samples in the batch, $x_i$ is the anchor sample, $x_i^+$ is the positive sample, $x_{i_j^-}$ is the $j$-th negative sample, $f(x, y)$ is the mapping function that projects samples $x$ and $y$ into the latent space, and $\mathcal{L}_{\text{SCL}}$ is the SCL loss.

**Disease Classification Loss**. Inspired by the approach of PromptMRG(Jin et al., 2024), an algorithm that adaptively adjusts learning objectives based on the learning states of different diseases, we introduce the logit-adjusted loss (Menon et al., 2020) to balance learning across diseases. This loss encourages the model to focus more on rare diseases by reducing their logits during optimization. For a given disease $D$, the logit-adjusted loss for the positive label $P$ is formulated as:

$$\mathcal{L}_{\text{SDL}}(y = P, f(\boldsymbol{x}^E)) = -\log \frac{e^{f_y(\boldsymbol{x}^E) + \log \pi_D}}{\sum_{y' \neq P} e^{f'_y(\boldsymbol{x}^E)} + \left(e^{f_y(\boldsymbol{x}^E) + \log \pi_D}\right)}. \tag{9}$$

**Total Training Loss.** The language modeling loss is used as the primary loss: $\mathcal{L}_{\mathrm{LM}} = -\sum_{t=1}^{T} \log p(r_t|r_1, ..., r_{t-1}, X, d_1, ..., d_L)$. The total training loss for our model is: $\mathcal{L} = \mathcal{L}_{\mathrm{LM}} + \lambda\mathcal{L}_{\mathrm{SDL}} + \gamma\mathcal{L}_{\mathrm{SCL}}$.

## 3. Experiments

### 3.1. Datasets and Metrics

**Datasets:** We validate the proposed method using two public datasets: MIMIC-CXR and IU X-Ray. **MIMIC-CXR** (Johnson et al., 2019) is currently the largest dataset containing chest X-ray images paired with corresponding textual reports. This dataset includes 377,110 chest X-ray images and 227,835 free-text radiology reports. Following the official split and the preprocessing steps, the resulting training, validation, and test sets contain 270,790, 2,130, and 3,858 samples, respectively. **IU X-Ray** (Demner-Fushman et al., 2016) is another commonly used public dataset. This dataset contains 7,470 X-ray images (including both frontal and lateral views) and 3,955 radiology reports. The dataset is divided into training, validation, and test sets in a 7:1:2 ratio. However, due to the limited number of positive samples for certain diseases, the original test split is not ideal for disease-specific evaluation. Therefore, we evaluate the entire IU X-Ray dataset using a model trained on the MIMIC-CXR training set (Jin et al., 2024).

**Evaluation metrics:** To evaluate model performance, we employ both Natural Language Generation (NLG) metrics and Clinical Efficacy (CE) metrics. **NLG metrics** include BLEU, METEOR, and ROUGE-L. Specifically, BLEU and METEOR were proposed for machine translation evaluation, while ROUGE-L is designed to assess the quality of summaries. **CE metrics** are used to evaluate the clinical validity of generated reports. We apply CheXBert(Smit et al., 2020) to tokenize the generated reports and compute precision, recall, and F1 scores based on the predicted labels.

**Implementation:** Our method employs an ImageNet-pretrained ResNet-101 model as the encoder and utilizes Bert-base as the decoder. The optimizer of choice is AdamW, with a weight decay rate set to 0.05. The initial learning rate is set to $5 \times 10^{-5}$ and dynamically adjusted using a cosine annealing strategy. In Eq. (12), $\lambda$ is dynamically adjusted during training, and $\gamma$ is set to 0.1. The model is trained for 10 epochs with a batch size of 16.

### 3.2. Results

We conduct a comprehensive comparison of our proposed model with state-of-the-art (SOTA) methods, including R2Gen (Chen et al., 2020), M2TR (Nooralahzadeh et al., 2021), MKSG (Yang et al., 2022), CliBert (Yan and Pei, 2022), M2KT (Yang et al., 2023), METrans (Wang et al., 2023), KiUT (Huang et al., 2023), DCL (Li et al., 2023),RGRG (Tanida et al., 2023),HSA (Zhu et al., 2024) as well as PromptMRG(Jin et al., 2024) and MAN(Shen et al., 2024). Additionally, we found that the experimental results of the X-RGen (Chen et al., 2024a) are quite good. However, due to differences in datasets and experimental settings, we did not make a comparison here.Table 1 presents the experimental results on the MIMIC-CXR and IU X-Ray datasets. From the table, it can be observed that our method achieves SOTA performance across all three Clinical Efficacy metrics on both datasets.Compared with the state-of-the-art PromptMRG framework, our method achieves

| Dataset | Model | Year | CE Metrics | | | NLG Metrics | | | |
|---|---|---|---|---|---|---|---|---|---|
| | | | Precision | Recall | F1 | BLEU-1 | BLEU-4 | METEOR | ROUGE-L |
| MIMIC | R2Gen | 2020 | 0.333 | 0.273 | 0.276 | 0.353 | 0.103 | 0.142 | 0.277 |
| | M2TR | 2021 | 0.240 | 0.428 | 0.308 | 0.378 | 0.107 | 0.145 | 0.272 |
| | MKSG | 2022 | 0.458 | 0.348 | 0.371 | 0.363 | 0.115 | - | 0.284 |
| | CliBert | 2022 | 0.397 | 0.435 | 0.415 | 0.383 | 0.106 | 0.144 | 0.275 |
| | M2KT | 2023 | 0.420 | 0.339 | 0.352 | 0.386 | 0.111 | - | 0.274 |
| | METrans. | 2023 | 0.364 | 0.309 | 0.311 | 0.386 | 0.124 | 0.152 | **0.291** |
| | KIUT | 2023 | 0.371 | 0.318 | 0.321 | 0.393 | 0.113 | 0.160 | 0.285 |
| | DCL | 2023 | 0.471 | 0.352 | 0.373 | - | 0.109 | 0.150 | 0.284 |
| | RGRG | 2023 | 0.461 | 0.475 | 0.447 | 0.373 | **0.126** | **0.168** | 0.264 |
| | MAN | 2024 | 0.411 | 0.398 | 0.389 | 0.396 | 0.115 | 0.151 | 0.274 |
| | HSA | 2024 | 0.480 | 0.357 | 0.379 | 0.386 | 0.120 | 0.163 | 0.288 |
| | PromptMRG | 2024 | 0.501 | 0.509 | 0.476 | 0.398 | 0.112 | 0.157 | 0.268 |
| | **DiffRGenNet (ours)** | - | **0.512** | **0.513** | **0.483** | **0.402** | 0.119 | 0.163 | 0.275 |
| IU X-Ray | R2Gen† | 2020 | 0.141 | 0.136 | 0.136 | 0.325 | 0.059 | 0.131 | 0.253 |
| | CVT2Dis.† | 2022 | 0.174 | 0.172 | 0.168 | 0.383 | 0.082 | 0.147 | 0.277 |
| | M2KT† | 2023 | 0.153 | 0.145 | 0.145 | 0.371 | 0.078 | 0.153 | 0.261 |
| | DCL† | 2023 | 0.168 | 0.167 | 0.162 | 0.354 | 0.074 | 0.152 | 0.267 |
| | RGRG† | 2023 | 0.183 | 0.187 | 0.180 | 0.266 | 0.063 | 0.146 | 0.180 |
| | PromptMRG† | 2024 | 0.213 | 0.229 | 0.211 | 0.401 | 0.098 | 0.160 | 0.281 |
| | **DiffRGenNet (ours)** | - | **0.216** | **0.230** | **0.213** | **0.417** | **0.104** | **0.167** | **0.309** |

Table 1: Comparison with MRG methods on MIMIC-CXR and IU X-Ray datasets. '†' indicates the performance evaluated by us. The best results are in bold.

a nearly 1-point improvement in CE metrics and a 3-point enhancement in ROUGE-L. This demonstrates the model's superior ability to capture positive samples and balance precision and recall. Additionally, learning from negative samples further enhances the model's ability to identify positive samples. However, there is still room for improvement in NLG metrics, particularly in generating long and complex sentences, which requires further optimization in future work.

### 3.3. Model Analysis

**Ablation study:** To validate the effectiveness of each module, we conduct ablative experiments on the MIMIC dataset. The results as in Table 2 indicate that removing the diff module leads to a slight performance degradation, while removing the contrastive learning module results in a significant performance drop, highlighting the critical role of diff negative sample learning in report generation. Additionally, the removal of the FAM module also causes a noticeable decline in performance. Overall, each module contributes positively to the MRG task, validating the rationality and necessity of their design.Additionally, the impact of the number of layers in the FAM module and the ablation study of various similarity metrics in the Feature Difference module are provided in Appendix A and Appendix C.

**Qualitative results:** We present a qualitative example to demonstrate the superiority of DiffRGenNet over the baseline. As in Figure 3, red text highlights key descriptions in the report, purple text indicates errors, and shaded text represents the differential changes of interest. Our method accurately generates a report consistent with the ground truth. It correctly assesses both normal and abnormal conditions, with particular attention to

| Model | B-1 | B-4 | M | R-L | F1 |
|---|---|---|---|---|---|
| DiffRGenNet (ours) | 0.402 | 0.119 | 0.163 | 0.275 | 0.483 |
| w/o Feature Difference | 0.401 | 0.115 | 0.160 | 0.269 | 0.481 |
| w/o FAM w Transformer | 0.400 | 0.112 | 0.159 | 0.268 | 0.477 |
| w/o SCL | 0.386 | 0.107 | 0.148 | 0.277 | 0.382 |

Table 2: Ablation study of each module on MIMIC dataset.

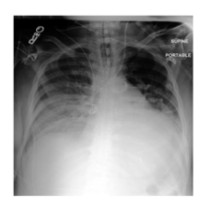

**Ground-Truth**

Comparison is made to the prior study performed two hours earlier. Interval placement of a nasogastric tube whose distal tip and sideport are below the gastroesophageal junction. Endotracheal tube and right IJ central line are in unchanged position. There is persistent cardiomegaly. There is a left retrocardiac opacity. There is prominence of the pulmonary vascular markings consistent with mild pulmonary edema. There is some atelectasis at the left lung base.

**PromptMRG**

As compared to the previous radiograph there is no relevant change. The monitoring and support devices are in constant position. The bilateral parenchymal opacities are constant in extent and severity. Also constant are the small pleural effusions and moderate cardiomegaly with mild-to-moderate pulmonary edema. Minimal atelectasis at the left lung bases.

**Ours**

Comparison to the prior study reveals stable positioning of monitoring devices. Persistent cardiomegaly is observed with pulmonary vascular congestion. A left retrocardiac opacity is present, consistent with mild pulmonary edema. Left basilar atelectasis remains evident.

Figure 3: Qualitative examples of the baseline (Jin et al., 2024) and the proposed method. Red indicates consistent content with the ground- truth while purple indicates incorrect one.

changes in abnormalities. For instance, the baseline method (Jin et al., 2024) incorrectly generates "the small pleural effusions," which is not present in the ground truth, and provides an imprecise description of "mild-to-moderate" for pulmonary edema. The Appendix presents more experimental results. Another Examle is provided in the Appendix B.

## 4. Conclusion

In this paper, we propose an effective method for medical report generation, DiffRGen-Net, designed to capture fine-grained features from both global and local dynamics, with a particular focus on regions of disease progression. We introduce the FAM (FlexiRoute Aggregation Module), which significantly enhances fine-grained feature extraction. Additionally, the proposed Diff module strengthens attention to areas of disease change. Finally, by employing contrastive learning with positive and negative samples, we further improve the model's generalization, robustness, and ability to identify rare diseases.

## Acknowledgments

Supported by Natural Science Foundation of China under Grant 62271465 and Suzhou Basic Research Program under Grant SYG202338.

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

## Appendix A. Variations in Dynamic Routing Settings

To explore the most suitable routing module, we conducted experiments on the MIMIC dataset with two types of routing modules: the impact of varying the number of routing layers in the FAM module and the influence of different routing architectures.

As in Figure A, we compared routing layers ranging from 1 to 5. The experimental results indicate that as the number of routing layers increases, the model's accuracy improves from 0.46 with 1 layer to 0.48 with 2 layers, but gradually decreases to 0.43 starting from 3 layers. Additionally, as illustrated in Table. A, we evaluated different routing architectures, including a standard Transformer-based architecture, a TRAR-based architecture, and the simplest approach of merely concatenating the two features without routing. The results demonstrate that our current routing architecture is the most suitable.

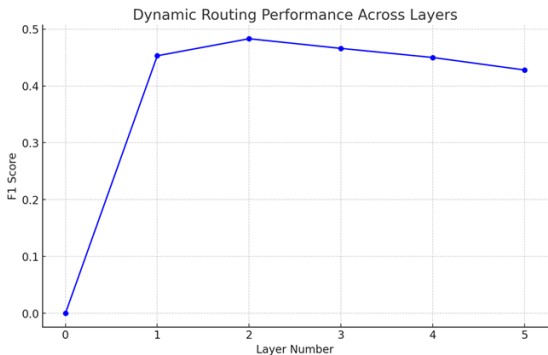

| Module | B-1 | B-4 | M | R-L | F1 |
|---|---|---|---|---|---|
| Transformer | 0.399 | 0.113 | 0.160 | 0.269 | 0.476 |
| TRAR | 0.400 | 0.116 | 0.161 | 0.272 | 0.478 |
| Concat | 0.313 | 0.101 | 0.140 | 0.265 | 0.267 |
| Ours(FAM) | 0.402 | 0.119 | 0.163 | 0.275 | 0.483 |

Table A: The effect of different routing architectures on the network.

Figure A: The effect of varying the number of routing layers in the FAM module on the network.

## Appendix B. Example of the model's ability to capture meaningful differences

We believe that the model's ability to capture differences lies in its capacity to detect variations in severity, as illustrated in Figure 4. Words highlighted in gray represent the severity of diseases, while those in red indicate the presence of diseases. For our method, it can be observed that terms indicating severity in the Ground-Truth are accurately captured, such as "moderate" and "mild," which are reflected as "mild to moderate," "subtle," and "early" in our approach. Additionally, the term "borderline" demonstrates our method's capability to capture more fine-grained changes.

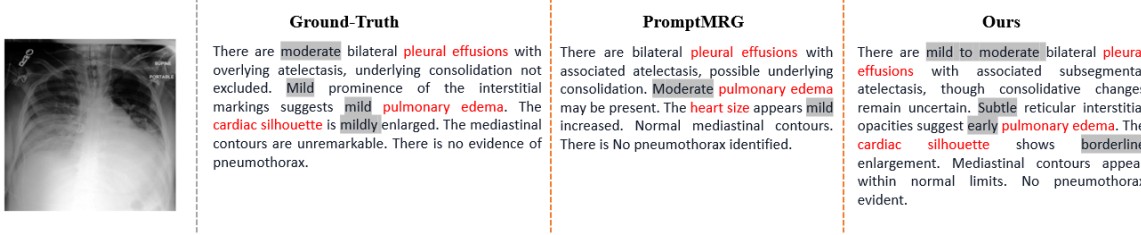

Figure 4: Qualitative examples of the baseline and the proposed method. Red indicates consistent content with the ground- truth while gray indicates the severity of diseases.

## Appendix C. Ablation Study of the Feature Difference Module

To validate the effectiveness of the three distinct metrics in the Feature Difference Module, we conducted three separate experiments to demonstrate: (1)Removing the L2 distance and dot product similarity. (2)Removing the dot product similarity and cosine similarity. (3)Removing the L2 distance and cosine similarity.

The ablation studies demonstrated the necessity of all three modules, and as shown in the table, the L2 distance is the most effective in capturing differences. The other two metrics also provide slight improvements.

| DiffRGenNet | B-4 | M | R-L | F1 |
|---|---|---|---|---|
| w/ Feature Diffrence | 0.119 | 0.163 | 0.275 | 0.483 |
| w/o L2 distance+dot product similarity | 0.115 | 0.160 | 0.270 | 0.481 |
| w/o dot product similarity+cosine similarity | **0.118** | **0.162** | 0.271 | **0.483** |
| w/o L2 distance+cosine similarity | 0.116 | 0.160 | **0.273** | 0.480 |

Table 3: Ablation study of Feature Difference Module on MIMIC dataset.

