# OpenReview forum: "DiffRGenNet: Difference-aware Medical Report Generation"
_MIDL.io/2025/Conference — MIDL 2025 Poster_

### Official Review · Reviewer_wwDZ · 2025-02-09

**Confidence:** 3
**Preliminary Rating:** 4
**Recommendation:** Poster

**Summary:**

The model retrieves **K** similar reports, then extracts **differential features** using multiple similarity metrics, including **L2 distance, cosine similarity, and dot product similarity**. To refine feature selection, it employs **Dynamic Routing Attention (DRA) layers**, which help balance **global and local dependencies**. A key component of the approach is **supervised contrastive learning (SCL)**

Experimental evaluations on the **MIMIC-CXR** and **IU X-Ray** datasets demonstrate that **DiffRGenNet outperforms state-of-the-art (SOTA) models**, including **R2Gen, DCL, and PromptMRG**, particularly in **Clinical Efficacy (CE) metrics**. Ablation studies further highlight the importance of key components, showing a **notable decline in F1-score** when removing the **Feature Difference Module** or **Contrastive Learning**. By effectively capturing **subtle disease progressions**, DiffRGenNet represents a **significant advancement** in **radiological AI**, helping reduce **radiologists' workload** while enhancing **diagnostic accuracy**.

**Strengths:**

The DiffRGenNet paper presents a technically robust and methodologically innovative approach to medical report generation (MRG) by focusing on fine-grained differences in radiological images. One of its standout strengths is its difference-aware framework, which enhances the detection of subtle disease progression—a critical factor in ensuring accurate and reliable clinical decision-making. Unlike conventional models that primarily emphasize disease classification, this work introduces a Feature Difference Module, specifically designed to highlight changes over time, making it particularly useful for longitudinal patient monitoring.

Another notable contribution is the integration of contrastive learning, which improves the model’s ability to distinguish between similar and dissimilar samples. This not only helps mitigate dataset biases but also enhances generalization to rare diseases. Additionally, the FlexiRoute Aggregation Module (FAM) dynamically selects global and local features, effectively addressing a common weakness in Transformer-based architectures, which often struggle to maintain a balance between fine details and overall context.

From a scientific perspective, the paper is well-structured and rigorously presented, following strong theoretical principles while providing extensive experimental validation on the MIMIC-CXR and IU X-Ray datasets. The authors employ both Natural Language Generation (NLG) and Clinical Efficacy (CE) metrics, ensuring a comprehensive evaluation of the model’s performance. While DiffRGenNet does not outright surpass all existing methods, it demonstrates strong clinical efficacy, underscoring its potential for real-world medical applications.

**Weaknesses:**

While DiffRGenNet presents a novel approach to medical report generation, there are several areas where improvements are needed.

First, the model’s advancements in Natural Language Generation (NLG) metrics are relatively modest compared to previous methods. Although it achieves state-of-the-art (SOTA) performance in clinical efficacy (CE) metrics, the authors acknowledge ongoing challenges with generating long and complex sentences. This suggests that while the model effectively captures disease-specific variations, it may still struggle with producing coherent and linguistically rich text, which is crucial for practical clinical applications.

Second, the evaluation methodology—while robust—is limited to just two datasets: MIMIC-CXR and IU X-Ray, both of which focus on chest X-ray reports. It remains unclear how well DiffRGenNet generalizes to other imaging modalities like MRI, CT scans, or ultrasound. Future work should expand validation efforts to a broader range of medical imaging datasets to ensure the model's applicability across diverse clinical scenarios.

Another potential drawback is the computational complexity introduced by the FlexiRoute Aggregation Module (FAM) and Feature Difference Module. While these components enhance fine-grained feature extraction, they also increase inference time and resource demands, which may hinder real-time clinical deployment. A detailed computational efficiency analysis comparing DiffRGenNet with simpler Transformer-based architectures would be valuable in assessing its practicality for clinical settings.

Additionally, the paper could provide a stronger comparison to prior research on differential feature extraction in medical imaging. Although contrastive learning and retrieval-based methods are well-studied in natural language processing (NLP) and computer vision, their specific application to medical report generation is still evolving. Clarifying how DiffRGenNet’s approach to contrastive learning differs from existing medical AI studies would strengthen its contribution.

Finally, while DiffRGenNet shows great promise, publicly available source code would significantly enhance its transparency and reproducibility. Releasing pretrained models and implementation details would facilitate validation and encourage broader adoption within the clinical AI research community.

**Detailed Comments:**

A Clear Introduction into the Structured
The Introduction

The introduction is an adequate motivation of the problem, but it would be better if it indicated more clearly how DiffRGenNet explicitly improves upon previous retrieval-based methods of medical report generation (for example knowledge graphs or memory-based retrieval). An overview table could sum up the main distinctions.

The high similarity among medical images is well-defined, but an example or some form of visualization showing that similarity from the dataset such as two similar X-rays with different diagnoses would add strength to the argument.
Methodology

Feature Difference Module is a very important module, but the functioning of the module is somewhat complicated. An understanding of the entire aspects would further improve clarity through a diagrammatic representation relating to the three similarity metrics - L2 distance, cosine similarity, and dot product similarity - and contribution of each of these similarity metrics in differential feature extraction.

The interest is in Gumbel-Softmax for routing probabilities in Equation 6, which is not justified enough. The authors should clarify why this was preferred over normal softmax or other attention-based selection models.
Importantly, Neg-Pos Matching by means of contrastive learning is an exciting complement; however, it must explain its newness in the domain of medical imaging. Materials by which previous works in self-supervised for medical imaging have used this kind of contrastive technique-how does it vary from these?
Experiments and Results

Although the MIMIC-CXR and IU X-Ray datasets are common to these research areas, the paper would also have benefitted from a discussion of dataset biases. Considering the bias in terms of diseases present within MIMIC-CXR-heavily skewed according to common diseases, how does the model deal with the representation of rare diseases?
The ablation study has been well structured but does not include efficiency comparison, such as inference time and memory usage, between DiffRGenNe.

**Justification Of The Preliminary Rating:**

Refer the review

While DiffRGenNet presents a novel approach to medical report generation, there are several areas where improvements are needed.

First, the model’s advancements in Natural Language Generation (NLG) metrics are relatively modest compared to previous methods. Although it achieves state-of-the-art (SOTA) performance in clinical efficacy (CE) metrics, the authors acknowledge ongoing challenges with generating long and complex sentences. This suggests that while the model effectively captures disease-specific variations, it may still struggle with producing coherent and linguistically rich text, which is crucial for practical clinical applications.

Second, the evaluation methodology—while robust—is limited to just two datasets: MIMIC-CXR and IU X-Ray, both of which focus on chest X-ray reports. It remains unclear how well DiffRGenNet generalizes to other imaging modalities like MRI, CT scans, or ultrasound. Future work should expand validation efforts to a broader range of medical imaging datasets to ensure the model's applicability across diverse clinical scenarios.

Another potential drawback is the computational complexity introduced by the FlexiRoute Aggregation Module (FAM) and Feature Difference Module. While these components enhance fine-grained feature extraction, they also increase inference time and resource demands, which may hinder real-time clinical deployment. A detailed computational efficiency analysis comparing DiffRGenNet with simpler Transformer-based architectures would be valuable in assessing its practicality for clinical settings.

Additionally, the paper could provide a stronger comparison to prior research on differential feature extraction in medical imaging. Although contrastive learning and retrieval-based methods are well-studied in natural language processing (NLP) and computer vision, their specific application to medical report generation is still evolving. Clarifying how DiffRGenNet’s approach to contrastive learning differs from existing medical AI studies would strengthen its contribution.

Finally, while DiffRGenNet shows great promise, publicly available source code would significantly enhance its transparency and reproducibility. Releasing pretrained models and implementation details would facilitate validation and encourage broader adoption within the clinical AI research community.

**Questions To Address In The Rebuttal:**

No

**Special Issue:**

No

---

> ### Author Response · Authors · 2025-03-08
>
> Thanks so much for the reviewer’s advice！I will reply to each of the proposed suggestions.
>
> >Comments:"First, the model’s advancements in Natural Language Generation (NLG) metrics are relatively modest compared to previous methods. Although it achieves state-of-the-art (SOTA) performance in clinical efficacy (CE) metrics, the authors acknowledge ongoing challenges with generating long and complex sentences. This suggests that while the model effectively captures disease-specific variations, it may still struggle with producing coherent and linguistically rich text, which is crucial for practical clinical applications."
>
> * Answer：In some examples, there are indeed issues with language coherence. However, in clinical practice, where there are numerous case samples, efficiency is crucial. Therefore, entity accuracy and clinical efficacy (CE) are particularly important for enabling doctors to quickly identify significant changes and diseases. Improving language coherence can be a focus for future optimization.
>
> >Comments:"Second, the evaluation methodology—while robust—is limited to just two datasets: MIMIC-CXR and IU X-Ray, both of which focus on chest X-ray reports. It remains unclear how well DiffRGenNet generalizes to other imaging modalities like MRI, CT scans, or ultrasound. Future work should expand validation efforts to a broader range of medical imaging datasets to ensure the model's applicability across diverse clinical scenarios."
>
> * Answer：hank you for the reviewers' suggestions. MIMIC and IU-Xray are currently the most commonly used datasets, so most work on MRG is conducted on these two datasets. In the future, we will consider transferring the model to other imaging modalities such as MRI, CT scans, or ultrasound to observe its effectiveness.
>
>
> >Comments:"Additionally, the paper could provide a stronger comparison to prior research on differential feature extraction in medical imaging. Although contrastive learning and retrieval-based methods are well-studied in natural language processing (NLP) and computer vision, their specific application to medical report generation is still evolving. Clarifying how DiffRGenNet’s approach to contrastive learning differs from existing medical AI studies would strengthen its contribution."
>
> * Answer：Existing contrastive learning strategies for selecting positive and negative samples typically rely on random or predefined methods to choose specific samples. We have adopted a dynamic sample selection strategy that selects the most suitable features after feature combination. Additionally, our approach uses supervised contrastive learning, with difference samples and similarity samples serving as supervision.
>
> >Comments:"Finally, while DiffRGenNet shows great promise, publicly available source code would significantly enhance its transparency and reproducibility. Releasing pretrained models and implementation details would facilitate validation and encourage broader adoption within the clinical AI research community."
>
> * Answer：We have released the code on GitHub.
>
>
> >Comments:"The introduction is an adequate motivation of the problem, but it would be better if it indicated more clearly how DiffRGenNet explicitly improves upon previous retrieval-based methods of medical report generation (for example knowledge graphs or memory-based retrieval). An overview table could sum up the main distinctions."
>
> * Answer：Thank you for the reviewers' suggestions. We have added an example demonstrating similarity, along with relevant figures, to the introduction. Therefore, we addressed this issue by including the text directly in the introduction.Our retrieval approach builds upon previous methods by utilizing the retrieved content in a more fine-grained manner. Previous methods would use the retrieved content in a rough and direct way, whereas we differentiate between similar and significantly different aspects of the retrieved content, thereby enhancing the learned knowledge.
>
> >Comments:"The high similarity among medical images is well-defined, but an example or some form of visualization showing that similarity from the dataset such as two similar X-rays with different diagnoses would add strength to the argument. Methodology"
>
> * Answer：Thank you for the reviewers' suggestions. We have added visualizations demonstrating similarity to the appendix.

---

> > ### Author Response · Authors · 2025-03-08
> >
> > >Comments:"Feature Difference Module is a very important module, but the functioning of the module is somewhat complicated. An understanding of the entire aspects would further improve clarity through a diagrammatic representation relating to the three similarity metrics - L2 distance, cosine similarity, and dot product similarity - and contribution of each of these similarity metrics in differential feature extraction."
> >
> > * Answer：We have added the ablation study for the Feature Difference Module to the appendix. The charts demonstrate the contributions of the three similarity metrics: L2 distance is the most important for the module, while the other two metrics also show their necessity.
> >
> > >Comments:"The interest is in Gumbel-Softmax for routing probabilities in Equation 6, which is not justified enough. The authors should clarify why this was preferred over normal softmax or other attention-based selection models. Importantly, Neg-Pos Matching by means of contrastive learning is an exciting complement; however, it must explain its newness in the domain of medical imaging. Materials by which previous works in self-supervised for medical imaging have used this kind of contrastive technique-how does it vary from these? Experiments and Results"
> >
> > * Answer：We use Gumbel-Softmax instead of standard softmax or attention mechanisms for routing probabilities. The core advantage of Gumbel-Softmax is its ability to achieve approximately discrete routing while maintaining differentiability. This approach addresses the gradient discontinuity problem caused by the non-differentiable argmax in standard softmax and dynamically balances stochastic exploration with deterministic allocation through a temperature annealing mechanism. It is more suitable for multi-task scenarios requiring hard routing.
> >
> > >Comments:"Although the MIMIC-CXR and IU X-Ray datasets are common to these research areas, the paper would also have benefitted from a discussion of dataset biases. Considering the bias in terms of diseases present within MIMIC-CXR-heavily skewed according to common diseases, how does the model deal with the representation of rare diseases? "
> >
> > * Answer：In the case of rare diseases, our model's Diff module can more effectively detect differences between rare diseases and common conditions, enhancing the distinction through learning from negative samples. Additionally, to balance the learning across different diseases, we introduced the logit adjustment loss. This loss function encourages the model to learn more about rare diseases by downscaling logits during the optimization process.
> >
> > >Comments:"The ablation study has been well structured but does not include efficiency comparison, such as inference time and memory usage, between DiffRGenNe."
> >
> > * Answer: Thank you for the reviewers' reminder. The FAM module treats self-attention as a fully connected graph feature update function and constructs different adjacency masks for the defined attention span. Consequently, the task of path selection in the module can be translated into a mask selection task, significantly reducing additional costs. As for additional experimental comparisons, due to time constraints, we were unable to conduct those experiments at this moment.

---

### Official Review · Reviewer_hpys · 2025-02-16

**Confidence:** 4
**Preliminary Rating:** 4
**Recommendation:** Poster

**Summary:**

This paper presents a novel Difference-aware Report Generation Network (DiffRGenNet) designed to improve medical report generation by addressing the challenges posed by high similarity among medical images from the same anatomical region and temporal variations in patient scans.The idea and motive of the article are good, but the writing and construction of the article are difficult to follow.

**Strengths:**

The paper introduces a difference-aware framework that explicitly considers image variations, an aspect often overlooked in traditional medical report generation models.
The proposed FlexiRoute Aggregation Module enables dynamic selection of routing paths, potentially improving adaptability across different cases.

**Weaknesses:**

The structure of the method presentation is not entirely aligned with Figure 1a/b/c, making it difficult to follow the proposed model's workflow.

Performance improvements on MIMIC-CXR appear to be marginal, raising concerns about the practical benefits of the proposed approach.

The qualitative experimental results are insufficient. The sample cases in Figure 2 do not clearly demonstrate the advantages of the method, and highlighted differences should be explicitly marked for better visualization.

**Detailed Comments:**

The methodology section should be restructured to ensure consistency with Figure 1a/b/c for better clarity.

A parameter study should be conducted to investigate the impact of setting 𝐾 as the maximum index for DRA layers.

The effectiveness of the three distinct metrics in the Feature Difference Module should be quantitatively verified. Are all three metrics necessary, or could a subset achieve similar results?

The explanation of "Retrieve 𝑘  features" in Figure 1a needs to be expanded—what features are being retrieved, and how do they contribute to the final report generation?

The CPC loss depicted in Figure 1 is not discussed in the experimental section. If it is a key component, an ablation study should be provided.

On the MIMIC-CXR dataset, the reported improvements seem relatively small. Further clarification is needed regarding the practical impact of these improvements. Additionally, results on IU-Xray with promptMRG should be reported.

The qualitative examples provided in Figure 2 do not sufficiently illustrate the method’s advantages. Highlighted differences should be clearly marked, and more cases should be included for a thorough comparison.

**Justification Of The Preliminary Rating:**

This paper introduces a novel Difference-aware Report Generation Network (DiffRGenNet) to improve medical report generation by explicitly modeling image differences using a Feature Diff module and dynamically routing dependencies via the FlexiRoute Aggregation Module. The proposed method is well-motivated and evaluated on MIMIC-CXR and IU X-Ray datasets, showing promising results.

**Questions To Address In The Rebuttal:**

How does the selection of K in the DRA layers impact model performance? Would different values of 𝐾 significantly change the results?

Can the authors provide empirical evidence supporting the necessity of all three metrics in the Feature Difference Module?

How does the model compare to promptMRG on IU-Xray? Is DiffRGenNet still superior under this setting?

What specific advantages does DiffRGenNet provide over baseline methods, given the relatively small performance gains on MIMIC-CXR?

Could additional visualizations be provided to better highlight the model’s ability to capture meaningful differences in medical images?

**Special Issue:**

No

---

> ### Author Response · Authors · 2025-03-08
>
> Thanks so much for the reviewer’s advice！I will reply to each of the proposed suggestions.
> >Comments:"The methodology section should be restructured to ensure consistency with Figure 1a/b/c for better clarity."
>
> * Answer：Thank you for suggestions. We have structured the content in our paper to be consistent with the images.
>
> >Comments:"How does the selection of K in the DRA layers impact model performance? Would different values of 𝐾 significantly change the results?"
>
> * Answer：Thank you for reminder. We had previously forgotten to include references to the appendix in the main text. Due to space constraints, this issue was addressed in the appendix. In the DRA layer, the choice of the K value does indeed affect model performance. The optimal K value is 2, with model performance improving initially as K increases and then decreasing afterward.
>
> >Comments:"Can the authors provide empirical evidence supporting the necessity of all three metrics in the Feature Difference Module?"
>
> * Answer：We performed ablation studies on the Feature Difference Module, and the results are provided in the appendix. The L2 Distance is the most effective at capturing differences, while the dot product similarity and cosine similarity also offer slight improvements in capturing differences, indicating their necessity.
>
> >Comments:"The explanation of "Retrieve 𝑘 features" in Figure 1a needs to be expanded—what features are being retrieved, and how do they contribute to the final report generation?"
>
> * Answer：We retrieve the top K reports that are most similar to the paired report. These K features are combined through the FAM module, ultimately selecting the features that are most similar to the report and those with significant differences, providing additional knowledge enhancement for subsequent report generation. There was an error in the diagram; the K-features should point to the Feature Difference module.
>
> >Comments:"The CPC loss depicted in Figure 1 is not discussed in the experimental section. If it is a key component, an ablation study should be provided."
>
> * Answer：Regarding the CPC loss described in the figure, it actually corresponds to the Supervised Contrastive Learning (SCL) loss. The name was taken from the paper "Contrastive Predictive Coding Based Feature for Automatic Speaker Verification." It has now been changed to the more easily understood SCL loss in the paper. In the ablation study, this experiment is labeled as w/o SCL.
>
> >Comments:"How does the model compare to promptMRG on IU-Xray? Is DiffRGenNet still superior under this setting?"
>
> * Answer：Thank you for the reviewers' reminder. Due to an oversight, we forgot to include this information. Our model's performance on the IU-Xray dataset still surpasses promptMRG in both CE and NLG metrics. The experimental results have now been added to the paper.
>
> >Comments:"What specific advantages does DiffRGenNet provide over baseline methods, given the relatively small performance gains on MIMIC-CXR?"
>
> * Answer：
> 1. The proposed method also aligns more closely with the diagnostic process of doctors by focusing on the parts where changes occur, i.e., areas showing differences. These are the regions where pathological changes are most likely to appear, enhancing the method's explainability.
> 2. Additionally, this method introduces a novel knowledge enhancement technique based on differences, which could be a feasible enhancement approach for other methods in the future.
>
> >Comments:"Could additional visualizations be provided to better highlight the model’s ability to capture meaningful differences in medical images?"
>
> * Answer：In the appendix of the paper, we have provided additional visualizations. By capturing differences, we refer to detecting fine-grained variations. Our method is effective at capturing words that indicate changes in severity, such as "mild" and "moderate," and is also capable of accurately identifying more nuanced changes like "borderline enlargement."

---

### Official Review · Reviewer_uD3o · 2025-02-22

**Confidence:** 4
**Preliminary Rating:** 3
**Recommendation:** Poster

**Summary:**

This paper addresses the challenge of automatically generating accurate radiological reports by capturing subtle differences in medical images, especially those from the same anatomical region or taken at different time points. The authors propose a novel network—DiffRGenNet—that emphasizes differential information to improve report precision, particularly in the context of rare diseases.

**Strengths:**

Key Contributions:
•	Dynamic Routing for Feature Aggregation:
DiffRGenNet introduces a FlexiRoute Aggregation Module (FAM) that dynamically balances global and local dependencies using multiple Dynamic Routing Attention (DRA) layers. These layers compute attention weights and routing probabilities to optimally combine features from the image and retrieved similar reports. This dynamic routing mechanism enables the model to select the most relevant features for accurate report generation.
•	Feature Difference Module:
To specifically capture fine-grained differences between images, the network includes a Feature Difference Module. It quantifies differences using three metrics—L2 distance, cosine similarity, and dot product similarity. These metrics are fused via a multi-layer perceptron (MLP), enhancing the network’s ability to identify critical variations that are essential for generating detailed reports.
•	Contrastive Learning with Neg-Pos Matching:
The paper further strengthens its approach with a Neg-Pos Matching module that employs supervised contrastive learning. This module aligns positive samples (similar features) while distinguishing negative samples (dissimilar features). Additionally, a disease classification loss is used to mitigate the imbalance caused by rare diseases, encouraging the model to focus on subtle yet clinically significant variations.

**Weaknesses:**

The paper lacks clear quantitative comparisons and recent benchmark evaluations, introduces irrelevant techniques without proper citations, suffers from unclear terminology and typos, features incomprehensible figures, and employs an overly complex architecture that challenges interpretability and debugging.

**Detailed Comments:**

Authors should show that the proposed methods outperform by how many percent in what metrics compared to the most powerful comparison methods. Why the works for varying viewpoints, which are not directly related, are introduced here? Did you use some technique or compare with them?
     The authors need to cite the papers about designing auxiliary enhancement in the introduction.
Every technique you used should be cited and clearly point out how they are related to your work here. I found many pieces seem to be missed.
Some typos and unclear statements need to be corrected like differential and similarity information. Retrive in the figure (??)
    Use the vector graph to draw figure 1, the figure 1 is not self-explainable, making this work hard to understand.
    Not clear why sometimes it is K-feature and K-layers at other times. Do they have the same meaning?
    The recently published paper (X-RGen) achieved relatively high performance on the BLEU and meteror and rouge metrics. The author has not compared with them. Could the authors explain why? Are the experiment settings different from them?
Chen, Qi, Yutong Xie, Biao Wu, Xiaomin Chen, James Ang, Minh-Son To, Xiaojun Chang, and Qi Wu. "Act Like a Radiologist: Radiology Report Generation across Anatomical Regions." In Proceedings of the Asian Conference on Computer Vision, pp. 1-17. 2024.
    DiffRGenNet’s architecture integrates several modules—the FlexiRoute Aggregation Module (FAM), Feature Difference Module, and Neg-Pos Matching—which increases its overall complexity. This multi-stage design, particularly with dynamic routing and multiple attention layers, may make the model harder to interpret and debug compared to simpler architectures.

**Justification Of The Preliminary Rating:**

The paper introduces an innovative approach—DiffRGenNet—that leverages dynamic routing, a feature difference module, and contrastive learning to generate precise radiological reports, especially for rare diseases. However, its impact is limited by insufficient quantitative comparisons, unclear terminology and figures, missing citations for key techniques, and an overly complex architecture that may hinder interpretability. These strengths and weaknesses justify a preliminary rating that recognizes the novelty of the approach while noting significant areas for improvement.

**Questions To Address In The Rebuttal:**

Authors should show that the proposed methods outperform by how many percent in what metrics compared to the most powerful comparison methods. Why the works for varying viewpoints, which are not directly related, are introduced here? Did you use some technique or compare with them?
     The authors need to cite the papers about designing auxiliary enhancement in the introduction.
Every technique you used should be cited and clearly point out how they are related to your work here. I found many pieces seem to be missed.
Some typos and unclear statements need to be corrected like differential and similarity information. Retrive in the figure (??)
    Use the vector graph to draw figure 1, the figure 1 is not self-explainable, making this work hard to understand.
    Not clear why sometimes it is K-feature and K-layers at other times. Do they have the same meaning?
    The recently published paper (X-RGen) achieved relatively high performance on the BLEU and meteror and rouge metrics. The author has not compared with them. Could the authors explain why? Are the experiment settings different from them?
Chen, Qi, Yutong Xie, Biao Wu, Xiaomin Chen, James Ang, Minh-Son To, Xiaojun Chang, and Qi Wu. "Act Like a Radiologist: Radiology Report Generation across Anatomical Regions." In Proceedings of the Asian Conference on Computer Vision, pp. 1-17. 2024.
    DiffRGenNet’s architecture integrates several modules—the FlexiRoute Aggregation Module (FAM), Feature Difference Module, and Neg-Pos Matching—which increases its overall complexity. This multi-stage design, particularly with dynamic routing and multiple attention layers, may make the model harder to interpret and debug compared to simpler architectures.

**Special Issue:**

No

---

> ### Author Response · Authors · 2025-03-08
>
> Thanks so much for the reviewer’s advice！I will reply to each of the proposed suggestions.
> >Comments:"Authors should show that the proposed methods outperform by how many percent in what metrics compared to the most powerful comparison methods. "
>
> * Answer：Thank you for suggestions. We have included the corresponding explanations in the paper.Compared with the state-of-the-art PromptMRG framework, our method achieves a nearly 1-point improvement in CE metrics and a 3-point enhancement in ROUGE-L.
>
> >Comments:"Why the works for varying viewpoints, which are not directly related, are introduced here? Did you use some technique or compare with them?"
>
> - Answer：
> 1. Introducing varying viewpoints is meant to highlight the main focus points in current natural image Difference Caption tasks. While their difference modules are noteworthy and can be referenced, our medical images do not require attention to viewpoint variations since they are captured from the same perspective. Therefore, we can adopt their difference modules without considering viewpoint changes.
> 2. Additionally, we have not compared our method with those used in natural image Difference Captioning tasks. Because there are significant differences between the two data domains, and the tasks themselves are somewhat different; natural image Difference Captioning involves inputting two images and outputting a brief description of their differences.
>
> >Comments:" The authors need to cite the papers about designing auxiliary enhancement in the introduction. Every technique you used should be cited and clearly point out how they are related to your work here. I found many pieces seem to be missed. "
>
> * Answer：Our work employs retrieval-augmented techniques to retrieve reports that are most similar to the images we aim to generate. These reports are then used to learn and compare differences and similarities. Information about the retrieval techniques has now been included in the paper.
>
> >Comments:"Some typos and unclear statements need to be corrected like differential and similarity information."
>
> * Answer：Sorry for this mistakes. The corrections have now been made in the paper.
>
> >Comments:"Retrive in the figure (??) Use the vector graph to draw figure 1, the figure 1 is not self-explainable, making this work hard to understand."
>
> * Answer：Figure 1 has been redrawn as a vector graphic, and minor errors such as 'Retrieve' have been corrected. To enhance clarity, we made two modifications that require explanation:
> 1. The first modification is the path from the image directly to FAM. In fact, it passes through the encoder on the right side to obtain features, which are then combined with the N-features into FAM. Due to congestion in the diagram, this was misrepresented during the initial drawing.
> 2. The second modification is the path from the report database to Feature Difference. It should actually be the path from the retrieved N-features to Feature Difference.
>
> >Comments:"Not clear why sometimes it is K-feature and K-layers at other times. Do they have the same meaning? "
>
> * Answer：Thank you for suggestions. We acknowledge that there was indeed some ambiguity in this area. The K in K-feature and the K in K-layer of FAM are unrelated. K-feature refers to the retrieved K features, which have now been changed to N in the paper. The K-layer in FAM refers to the number of layers in the module.
>
> >Comments:"The recently published paper (X-RGen) achieved relatively high performance on the BLEU and meteror and rouge metrics. The author has not compared with them. Could the authors explain why? Are the experiment settings different from them?"
>
> * Answer：Thank you for providing the paper.
> 1. We found that this paper uses a privately constructed multi-site dataset and is trained in a specialised setting. Our method is conducted on public datasets and follows the same experimental setup as existing methods. Therefore, comparing our work with theirs would be unfair and infeasible.
> 2. We will cite this paper in our manuscript and include a discussion about it.

---

> > ### Author Response · Authors · 2025-03-08
> >
> > >Comments:"DiffRGenNet’s architecture integrates several modules—the FlexiRoute Aggregation Module (FAM), Feature Difference Module, and Neg-Pos Matching—which increases its overall complexity. This multi-stage design, particularly with dynamic routing and multiple attention layers, may make the model harder to interpret and debug compared to simpler architectures."
> >
> > * Answer：
> > 1. Interpretation: The proposed method aligns more closely with the diagnostic process of doctors by focusing on the parts where changes occur, i.e., areas showing differences. These are the regions where pathological changes are most likely to appear, enhancing the method's interpretation.
> > 2. Debugging: The code has been open-sourced, making it friendly for debugging. The FAM module treats self-attention as a fully connected graph feature update function and constructs different adjacency masks for the defined attention span. Consequently, the task of path selection in the module can be translated into a mask selection task, significantly reducing additional costs.

---

### Author Rebuttal · Authors · 2025-03-08

**Rebuttal:**

We have made revisions based on the reviewers' suggestions. The detailed discussion with the reviewer is included in the official comments, and the revised paper has also been submitted. Some of the modifications in paper are as follows:

* Introduction: Added examples of similar reports , mentioned our use of retrieval-enhanced techniques, and clarified how our approach differs from other retrieval techniques.

* Method: Adjusted the structure by swapping the FAM and Feature Difference modules to align with the process depicted in the images. Provided a more detailed explanation of the processes for these two modules. We also made some adjustments to the method diagram.

* Experiments: Added more comparative models in the experiments section.

* Appendix: Included two additional parts: visual examples of the capability to capture difference and ablation studies for the three metrics of the Feature Difference module.

**Supporting Material:**

/attachment/a70a42f87737727bfa648afe8b06d4f2298c1366.pdf

---

### Meta-Review · Area_Chair_HJx5 · 2025-03-19

**Recommendation:** Accept (Poster)
**Confidence:** 4

**Metareview:**

All reviewers found the proposed method to be novel and the results promising.